# Improved Stable Read Range of the RFID Tag Using Slot Apertures and Capacitive Gaps for Outdoor Localization Applications

**DOI:** 10.3390/mi14071364

**Published:** 2023-06-30

**Authors:** Redouane Jouali, Hassan Ouahmane, Jalal Khan, Maryam Liaqat, Azize Bhaij, Sarosh Ahmad, Abderrahim Haddad, Mohssin Aoutoul

**Affiliations:** 1LTI Laboratory, Ecole Nationale des Sciences Appliquées, Chouaib Doukkali University, El Jadida 24000, Morocco; red1jouali05@gmail.com (R.J.); ouahmane.h@ucd.ac.ma (H.O.); 2Department of Telecommunication Engineering, University of Engineering & Technology, Mardan 23200, Pakistan; jalal@uetmardan.edu.pk; 3Department of Physics, University of Okara, Okara 56300, Pakistan; 4STIC Laboratory, Faculty of Science, Chouaib Doukkali University, El Jadida 24000, Morocco; azize.bhaij@gmail.com (A.B.); haddaddoctorat@gmail.com (A.H.); mohssin.aoutoul@gmail.com (M.A.); 5Department of Signal Theory and Communications, Universidad Carlos III de Madrid, 28911 Leganes, Spain

**Keywords:** RFID system, UHF frequency band, read range, localization

## Abstract

This paper proposes a small-size UHF RFID tag antenna, which was designed to function in the frequency interval of 860–960 MHz, with a large-read range of up to 17 m. In this work, the effects of capacitive slots and gaps on the impedance matching between conventional industrial chips and a designed RFID antenna was investigated. Simulated and measured results provided a clear indication that these two techniques can efficiently improve the return loss parameter and the antenna impedance matching behavior at the UHF band. Three-dimensional electromagnetic (EM) simulations results further proved that a better impedance matching between an industrial chip and a proposed RFID antenna occurs at 870 MHz, where the estimated input antenna impedance was about *Z_a_* = 16 + *j*184 (Ω), and the calculated read range reached a value of up to 17 m with a measured return loss value of –14 dB. The proposed RFID antenna can almost maintain the same read range value over a 180 degree angle variations on the horizontal plane owing to its omnidirectional radiation pattern. The fabrication and test stages of the antenna prototype were scheduled to validate the simulated characteristics. Experimental results confirmed the performances of our proposed RFID tag antenna and proved its potential ability for localization applications. EM simulations have been performed using the well-known commercial EM software simulator CST MWS.

## 1. Introduction

Radio frequency identification (RFID) is a promising alternative solution for indoor and outdoor localization, and for the tracking of objects. Though satellite navigation systems, such as GPS and ground base stations have long been used for surveillance, navigation, localization, and timing services, they present their limitations in indoor environments or in particular outdoor areas, including in tunnels and dense forests, or during inclement weather. To remedy these deficiencies, solutions deploying embedded sensors [1,2] have been proposed to reinforce the GPS or GSM signal in order to exploit it for indoor positioning within large buildings; inertial navigation [3,4,5], performed by motion sensors (e.g., accelerometers and gyroscopes), was used to determine the position, orientation, and speed of vehicles, especially in “blind” areas where the GPS signal is undetectable.

During the last decade, there has been a special interest in outdoor localization assisted by active RFID tags using a multilateration technique (ranging), which consists of determining the position of an object by knowing the distances separating it from at least three fixed stations. These applications are intended primarily for rescue interventions during natural disasters to locate victims and objects [6,7,8,9]. Active RFID tags deployed do not have a high power consumption, meaning their batteries can remain for several months, and are located at a distance of a few hundred meters [6]. Furthermore, passive RFID tags, distinguished by their short communication distances (of a few meters maximum), are of particular interest for auto-localization or navigation applications. 

The localization of vehicles on the roads remains an attractive research topic involving passive tags for outdoor localization. Localization using passive tags can be performed either by relying on known techniques, such as fingerprinting and multilateration (ranging), both of which are based on measurements of the “Received Signal Strength” (RSS) returned by these passive tags followed by a calculation [10,11] to determine the position of the moving object, or by considering that the position of the object is indicated directly by the coordinates of the tag in the case where the object is very close to the RFID tag, as for vehicle localization applications using tags stuck on the roads [12,13].

RFID systems are competitive candidates for localization aims. They benefit from their radiated electromagnetic power to detect and track, automatically, and are used in the form of tags attached to objects [14]. RFID systems have gained increasing attention across different application areas, such as in transport, manufacturing, and supply systems [15,16,17]. The main UHF RFID band covers frequencies from 860 MHz to 960 MHz, respectively. However, depending on countries, just sub-intervals of the UHF frequencies are exploited. For example, in Europe, America/Canada, Korea, and Japan, their frequencies bands are 865–868 MHz, 902–928 MHz, 908.5–914 MHz, and 950–956 MHz, respectively [18,19]. An RFID system is based on two key components: a transponder and a “tag”. RFIDs have many advantages compared to other identification techniques, such as having a better accuracy, being very easy to be integrated within any object, and their ability to perform long-range identification. The only reason RFID tags have not yet replaced the barcode entirely is due to their manufacturing cost, which remains relatively expensive compared to the cost of the barcode [20].

Tag RFID antennas can be classified in two categories: passive and active tags. The latter ones contain an onboard battery as a power supply, which is not available in the case of passive RFID tags, meaning they harvest electromagnetic power when they are illuminated by the RF energy of a reader. Inside the tag, the RFID antenna transmits this energy to the chip [21]. Main performances of the tag depend on the impedance matching level between the antenna and the chip, and when the antenna is well matched with the chip, i.e., its input impedance is equal to the complex conjugate basis, it receives the maximum amount of power from the chip and vice versa [22,23]. This work aims to assess the effects of various geometrical parameters to maximize their tag range. These parameters concern the dimensions of a slot and two gaps of a designed UHF-RFID tag structure, where the antenna is supposed to be connected to an industrial Impinj M700 [24]. Several structure components can be investigated, but our study will focus more on the effect of the gaps and slot on the impedance matching process in order to improve the return loss parameter values, thereby obtaining a long read range RFID tag.

Impedance matching is a critical aspect of the antennas design and in particular RFID structures, as it ensures an efficient power transfer is made between the tag antenna and the chip, and as a result, it improves communication between the tag and the reader. Generally, a good antenna impedance matching requires the integration of additional microstrip components. In the literature, interesting techniques have been introduced for this purpose, like microstrip meandering lines, stubs, capacitive gaps, and etched slots. The antenna impedance can be affected by the capacitive effect, which is created by the gaps introduced between the microstrip line components [25,26]. Other techniques that are also included which we can cite as examples include T-matching, nested-slots, and gap loading.

T-matching involves using a T-shaped network, which is composed of resistors, capacitors, and inductors to adjust the impedance of the antenna to match the impedance of the circuitry to which it is connected. The nested-slot structure uses a rectangular patch antenna with the nested slots cut out of the metal patch to achieve a high level of impedance matching over a wide frequency range. Gap loading involves introducing a small gap into the antenna structure to adjust the antenna’s impedance [25,27].

In this work, we focused specifically on the effect of introducing gaps and slots in the design of a UHF-RFID tag antenna structure. The aim of this study was to determine how the dimensions of the slots and gaps can affect the impedance matching process and maximize the tag range. This study covers the effects of engraved slots and gaps, and specifically focuses on the positioning of the gaps for optimal performance. Impedance matching is a critical aspect of the antenna’s design, and the introduction of gaps and slots can provide an effective means of achieving it.

This manuscript is organized as follows: in Section 2, we discuss a few fundamentals of the RFID systems, such as impedances and the antenna read range. Section 3 introduces the designed optimal antenna, and shows the results of the parametric study, which covers the engraved slots and gaps. In Section 4, the experimental results are presented to validate the numerical study. Finally, Section 5 provides the conclusion.

## 2. The RFID System

### 2.1. The RFID Tag

In an RFID system, the chip can be modeled by an equivalent RC electrical circuit where the two components can either be in parallel or in series. Table 1 presents characteristics of some known used industrial chips.

Figure 1 presents the model of an RFID tag where the antenna is connected to a chip. The chip impedance, *Z_out_*, is equivalent to a resistor and a capacity in parallel (*R* = 2.37 KΩ, *C* = 0.94 pF) as shown in Figure 2. Its expression, in terms of frequency, is given by Formula (1) below. At 870 MHz, the chip imaginary part’s impedance is about −194 Ω. To obtain a good matching impedance and, hence, to increase the amount of energy received by the antenna, the impedance, *Z_in_*, should be the conjugate of the chip’s impedance, as mentioned previously.
(1)$$Z_{chip}=\frac{R\times \frac{1}{jC\omega }}{R+\frac{1}{jC\omega }}=\frac{R}{R^{2}C^{2}\omega^{2}-1}-j\frac{R^{2}C\omega }{R^{2}C^{2}\omega^{2}-1}\approx \;\frac{1}{RC^{2}\omega^{2}}-j\frac{1}{C\omega }=R^{\prime }+Z_{c}$$

### 2.2. The RFID Tag Read Range

When the antenna is well matched to the chip, which is the source of energy, more power can propagate in the surrounding medium through the aerial structure. This improves the long-read distance of the tag, which is a key element of RFID systems oriented to localization applications. Based on the Friis free space equation [29], the read range of a receiving RFID tag is expressed as follows:(2)$$d=\frac{c}{4\pi f}\sqrt{\frac{P_{t}\;G_{t}G_{r}\tau }{P_{th}}},$$
where *P_th_* is the chip power sensibility, *P_t_* is transmitted power from the reader, *G_t_* and *G_r_* are the RFID and reader antennas gains, respectively, and *τ* is the ratio of transmitted energy to the chip to that received by the antenna. In the receiving mode, *τ* can have the following expression in terms of the chip reflection parameter Г:(3)$$\tau =1-{Г}^{2},$$
(4)$$Г=\frac{Z_{chip}-Z_{antenna}}{Z_{chip}+Z_{antenna}}.$$

### 2.3. Antenna Design

The proposed tag antenna, as shown in Figure 3, is printed on a lossy Rogers material substrate having a relative permittivity *ε_r_* = 3.5, a loss tangent *δ* = 0.0013, and a thickness *h* = 2 mm, respectively. Geometrical parameters of the simulated design are summarized in Table 2.

## 3. Simulated Results and Discussion

### 3.1. Capacitive Gap Effect Study

In general, adding gaps to the antenna layout consists of introducing two metal surfaces separated by a distance *l_g_*. In the presence of these gaps, various parameters, such as their position, length, and width can significantly affect the antenna’s performance. Therefore, it is essential to understand the effects of these parameters on the antenna impedance, and how they can be optimized to improve its performance.

To obtain more knowledge about their influence on the antenna’s performance, we will explore in more depth in the next paragraphs the impacts of these parameters through a series of parametric studies, and how they can be adjusted to achieve optimal characteristics in the presence of gaps or/and slots.

It should be noted that all 3D EM simulations, whose results are presented in this paper were performed using a 50 Ω discreet port excitation.

#### 3.1.1. Gaps Length Effect on the Return Loss Parameter

The first simulation concerned the effect of creating gaps on the proposed structure, as shown in Figure 3. Results depicted in Figure 4 show that the gap dimension significantly affects the performance and characteristics of the proposed RFID antenna, since the operating frequency was found to have shifted to the upper values, while the return loss parameters lowered to below −10 dB. Moreover, when the gap length increases, the reflection coefficient at resonant frequency slightly dropped as a result.

Figure 5 shows the effect of the gap length on the impedance of the proposed antenna. Without the gaps, the antenna impedance values were low compared to the chip impedance, resulting in a bad impedance matching between the antenna and the chip. As the gap length increased, the impedance peak values decreased, and the highest values of the impedance, including both the imaginary part and the real part, were localized at the upper values of these frequencies. Based on the chip impedance, *Z_c_* = 16 − *j*184 (Ω), the antenna impedance should have a high imaginary value and low real value, respectively. Hence, the optimal value of the gap length at the UHF band will be *l_g_* = 2 mm, as it offers the nearest antenna impedance, *Z_a_* = 28 + *j*184 (Ω) at 870.3 MHz, to the complex conjugate of the chip impedance.

#### 3.1.2. Gap Position Effects on the Antenna Impedance

The gap length is not the only factor that can alter the antenna performance. In fact, the gap position (Figure 6) also presents a key parameter that can affect the antenna’s characteristics. Based on the author’s knowledge, there was no study which considered the gap position effect in the literature. Furthermore, the authors observed that a change in the gap position led to a significant shift on the antenna impedance, as depicted in Figure 7.

A numerical study was performed in order to determine the effects on antenna impedance created by the gaps on two selected positions shown in Figure 6. The outcome results of this study are presented in Figure 7, which verifies the key role of the gap position. In fact, position 1 provides a good antenna impedance matching since the obtained antenna impedance, *Z_a_* = 28 + *j*184 (Ω), is very close to the complex conjugate impedance of the chip, whereas position 2 provides an antenna impedance *Z_a_* = 3 + *j*68 (Ω), which leads to a significant mismatching, and hence the transmitted amount of power between the chip and the RFID antenna will decrease.

It should be noted that the aim of the last simulation studies was to highlight an important factor that should be considered in the RFID antenna design. Thus, there is a potential chance to obtain better results when investigating in other positions and length values of these gaps. As a conclusion, the created gaps significantly helped in improving the performance of this antenna structure by affecting its impedance. However, this last quantity can be more refined, especially its real part, using other techniques like slots, as will be discussed in the coming section.

### 3.2. Rectangular Slotted Aperture

Theoretically, the insertion of slots in a radiating structure is often used to disrupt the circulation of currents on the top metal layer of the microstrip structure. The current will be forced to bypass these slots and will therefore have to take a longer path than the path without the slots [30]. The insertion of these slots induces capacitive and inductive effects, which thereby make it possible to modify the input impedance of the antenna. Hence, the aim of this subsection was to assess how the slots dimensions, both length and width, can affect the antenna impedance.

#### 3.2.1. Effect of the Slot’s Length

The first simulation concerns only the effect of the slot length on the antenna impedance. This study was performed using the antenna structure with the optimal gap length, *l_g_* = 2 mm. Figure 8 presents the reflection coefficient (*S*_11_) of the antenna port for the different sizes of *l_sl_*. According to the graph, the slot length had a minimal effect on the return loss.

Figure 7 presents the antenna impedance variations in terms of the frequency versus the different slot length values. These results show that the real and the imaginary parts decrease as the slot length increases.

Based on the comparison of antenna impedance for both the real part and the imaginary part, it can be determined that while the length of the slot decreases, the real part decreases, while the imaginary part increases slightly. 

The proposed gap antenna has an impedance *Z_a_* = 28 − *j*186. The goal of this section was to configure how to decrease the real part to obtain a well-matched antenna impedance to that of the chip. We can confirm that the engraved slot can decrease the real part, with its optimal value having been located at around 870 MHz, *X_a_* = 16 Ohm, when the slot dimensions values were as follows: *l_sl_* = 14 m and *W_sl_* = 2 mm, respectively. The imaginary part was found to change slightly in this context, as illustrated in Figure 9b.

Figure 9 shows the effect of the slot width on the antenna impedance of the proposed antenna structure. As observed in Figure 9, slight changes were found to have occurred when changing the slot width of the imaginary part of the antenna impedance. 

#### 3.2.2. Effect of the Slot’s Width

In order to investigate how the slot width affects the performance of the antenna, a numerical study was conducted comparing the proposed antenna versus different slot width values, *W_sl_* = 0, 1, 2 mm.

Figure 10 presents the reflection coefficient values, *S*_11_, at the antenna port versus different values of the slot width, *W_sl_*. Based on these plotted curves, it was immediately concluded that both the resonant frequency and the bandwidth changed slightly when the *W_sl_* value was increased. Moreover, the return loss at the resonant frequency decreased while the width of the slot became larger. Another effect was indicated in Table 3, whereby the engraved slot can reduce the antenna impedance to both of its real and imaginary parts. The optimal antenna impedance was found to be 16 + 184*j*, when *W_sl_* = 2 mm, which is nearly the complex conjugate of the chip impedance.

## 4. Optimal Structure Performances

According to the conducted simulation, the antenna-matching impedance was optimized when creating a gap of length 2 mm, and slots of dimensions *W_sl_* = 2 mm and *L_sl_* = 14 mm, respectively. For further investigations, other geometrical parameters can be optimized to validate the performances of our proposed RFID structure.

### 4.1. Input Antenna–Impedance Matching

The parametric study shown in Figure 8 proves that the length of the gaps and the dimensions of the slot play a decisive role in satisfying the impedance matching criteria, whereby the input impedance of the tag antenna nearly corresponds to the complex conjugate impedance of the chip, *Z_a_* = *Z*_chip_*, around 870 MHz as illustrated by Figure 11, and is equal to *Z_a_* = 16.90 + *j*184.88 (Ω), indicating that there is a satisfying level of impedance matching between the RFID antenna and the chip.

### 4.2. Read Rang and Return Loss

Based on the Friis Equation (1), the *S*_11_ values, and the radiation pattern results at 870 MHz, the read range parameters were calculated as shown in Table 4, taking into account the effects of the gaps and the slot.

It can be seen from Table 4 that a satisfactory impedance matching level was obtained at the simulated resonant frequency with significant improvements in the read range achieved due to the gap and slot effects. The antenna complex conjugate impedance has been adjusted to fit that of the commercially available tag chip Impinj M730, which has a sensitivity of −24 dBm and an output impedance equal to *Z_c_* = 15 − *j*189 (Ω) at 870.3 MHz, respectively. The performance of designed tag antenna has been assessed by analyzing its reading range and return loss over a wide UHF frequency band. 

To evaluate the response of the antenna in terms of its return loss, when connecting it to the chip, the *S*_11_ parameter of the designed RFID tag was recalculated using the output impedance of the industrial chip Impinj M730 and the already calculated *S*_11_ during the previous 3D EM simulation process (the results of which are shown in Figure 8 and Figure 10). Figure 12 shows clearly that the resonant frequency of the UHF RFID tag antenna is around 870 MHz with a return loss of less than −10 dB, whereas the 3D EM simulations illustrated a resonant frequency around 890 MHz, respectively.

Table 5 below presents a comparison in terms of the size, bandwidth, and read range parameters between our proposed RFID antenna and that of previously reported similar structures in the literature. From examining these results, it seems that our proposed antenna is significantly smaller and offers a large read range at the UHF frequency band.

## 5. Experimental Results

A prototype of the simulated RFID tag antenna was fabricated, as shown in Figure 13, using a Rogers substrate of a permittivity of 3.5 and a thickness of 2 mm, respectively. The fabricated structure has a size of 95 mm × 25 mm, with two gaps of 2 mm as the length, an etched slot of 2 mm as the width, with 14 mm as the length. Figure 14 presents the *S*_11_ parameter variations of the designed antenna in both the simulation and measurement cases, with and without considering the gaps and the slot. Results depicted in Figure 14 show the key role of the introduced gaps and the engraved slot, since the structure became extremely reflective of the coming power when there were no gaps and slots. The radiation pattern is considered as extremely important in localization application since the calculated read range parameter is based on it. Figure 15 and Figure 16 show a dipole-like antenna case of the radiation pattern in the XY plane, and hence the read range of the proposed RFID antenna will be almost the same over a 180 degree angle variations. Though radiation values of the two curves shown in Figure 15b are comparable, the diagram of the case “no gaps and no slot” does not reflect a real EM radiation since the relevant return loss values are closer to 0 dB, as illustrated in Figure 14, meaning that no power can be delivered to the antenna in the case of the absence of the etched gaps and slot. Notably, we should mention that the gain parameter is a ratio between the accepted power coming from the source and the radiated portion of this power by the antenna. The experimental results, depicted in Figure 14 and Figure 15, are in a good agreement with the calculated ones, which validates our 3D EM simulation process. We have to note that the higher simulated value of radiation efficiency was recorded at around 68% at 900 MHz.

## 6. Conclusions

This work presents a new design of an UHF RFID tag antenna for localization applications. The proposed optimal antenna structure efficiently exploits the capacitive effect induced using two techniques, namely etched gaps and slots, to achieve a good impedance matching between the RFID antenna and the chip. Engraved gaps is an interesting technique that has been used to create capacitive effects inside the antenna, where its effectiveness depends on the length and the position of the gaps. The second technique uses a rectangular slot to refine and enhance the antenna–chip impedance matching, which is influenced by the slot dimensions. The determination of the gaps’ and slots’ optimal dimensions can be achieved through an iterative process of the EM simulations, during which the real and imaginary parts of the antenna impedance have been refined to match the chip impedance.

Both the simulated and experimental results validated the performances of the proposed RFID antenna. The optimal design presents interesting results with a measured return loss value about −14 dB at 891.7 MHz, along with a satisfactory simulated impedance matching having been achieved at around 870 MHz in cases where the antenna was connected to the chip in these simulations. Furthermore, our proposed antenna stands out with a long read range of 17 m over an approximate 180 degree angle variations in the horizontal plane, which makes it a competitive candidate for vehicle localization applications.

## Figures and Tables

**Figure 1 micromachines-14-01364-f001:**
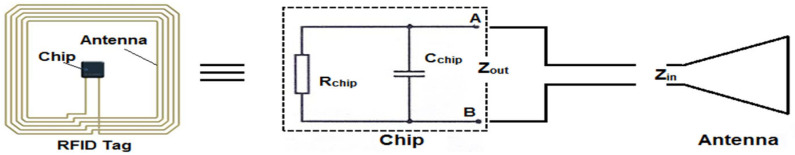
The model UHF RFID tag.

**Figure 2 micromachines-14-01364-f002:**
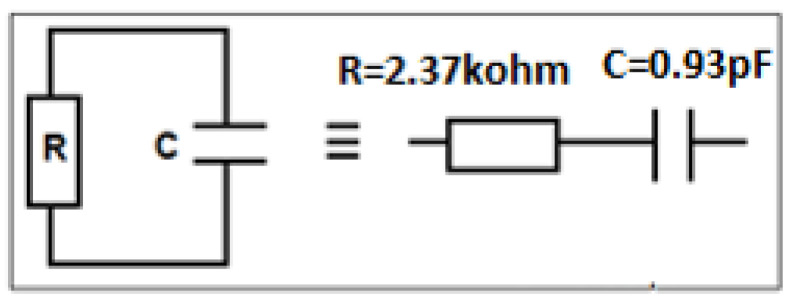
Model of the chip Impinj M730 [24].

**Figure 3 micromachines-14-01364-f003:**
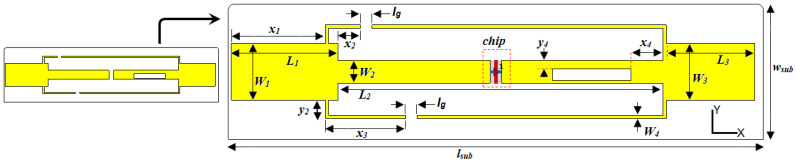
Proposed RFID antenna structure and its geometrical parameters in detail.

**Figure 4 micromachines-14-01364-f004:**
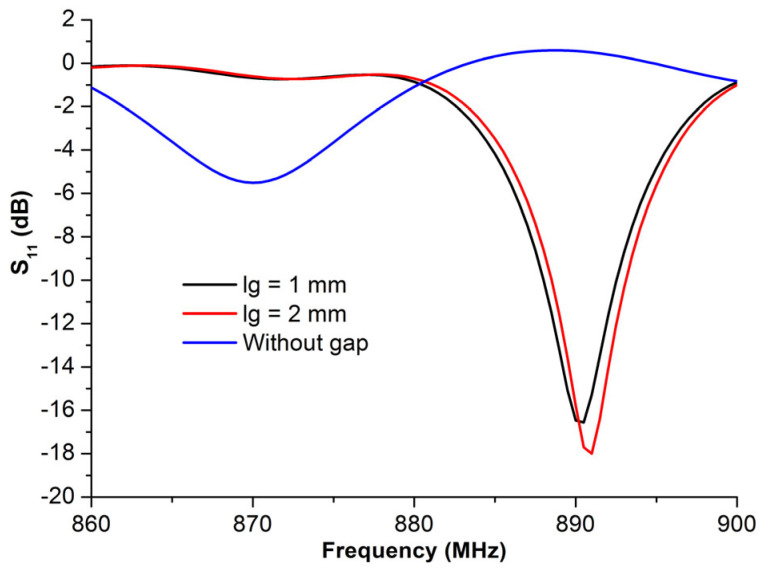
*S*_11_ with different gap lengths (*l_g_*).

**Figure 5 micromachines-14-01364-f005:**
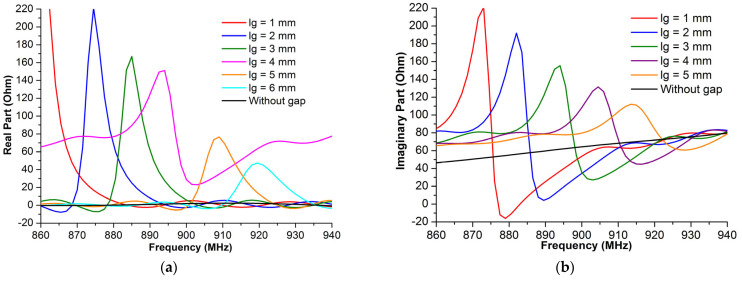
The proposed RFID antenna impedance vs. different gap lengths *l_g_*: (**a**) real part and (**b**) imaginary part.

**Figure 6 micromachines-14-01364-f006:**
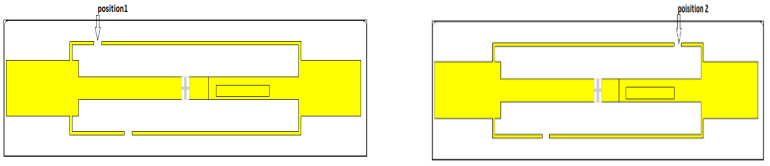
RFID antenna structure with two considered gap positions.

**Figure 7 micromachines-14-01364-f007:**
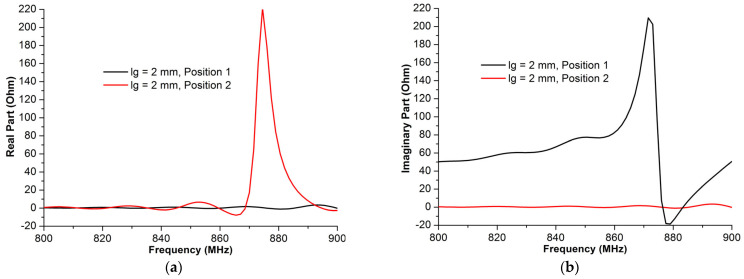
The antenna impedance, real part (**a**) and imaginary part (**b**), of the proposed structure in a different gap position.

**Figure 8 micromachines-14-01364-f008:**
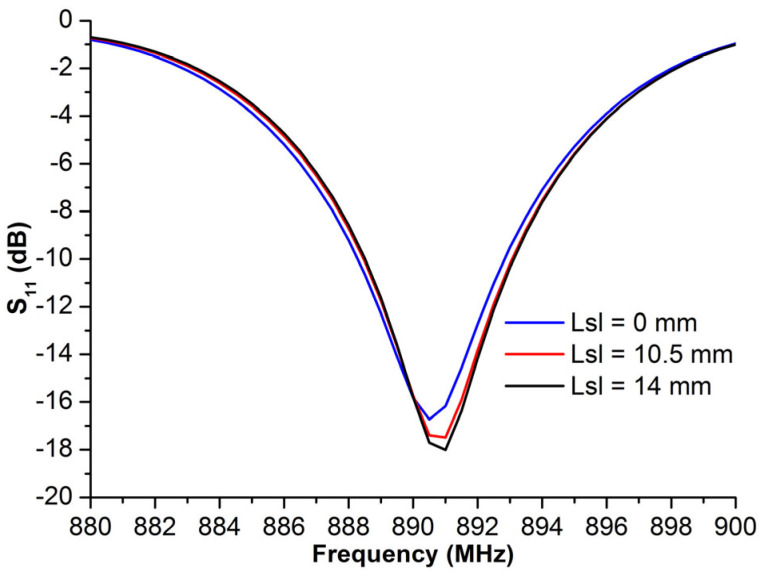
Return loss, *S*_11_, of the proposed structure vs. the different slot length, *l_sl_*.

**Figure 9 micromachines-14-01364-f009:**
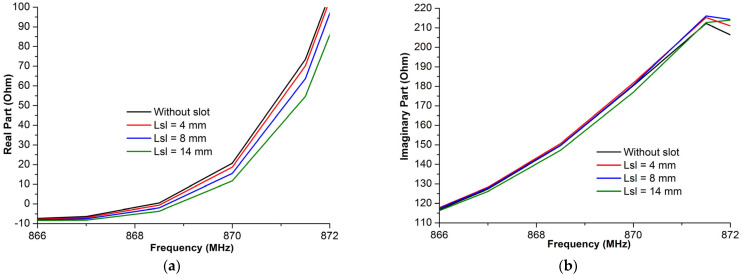
The antenna impedance, real part (**a**), and imaginary part (**b**) of the proposed structure vs. the slot length.

**Figure 10 micromachines-14-01364-f010:**
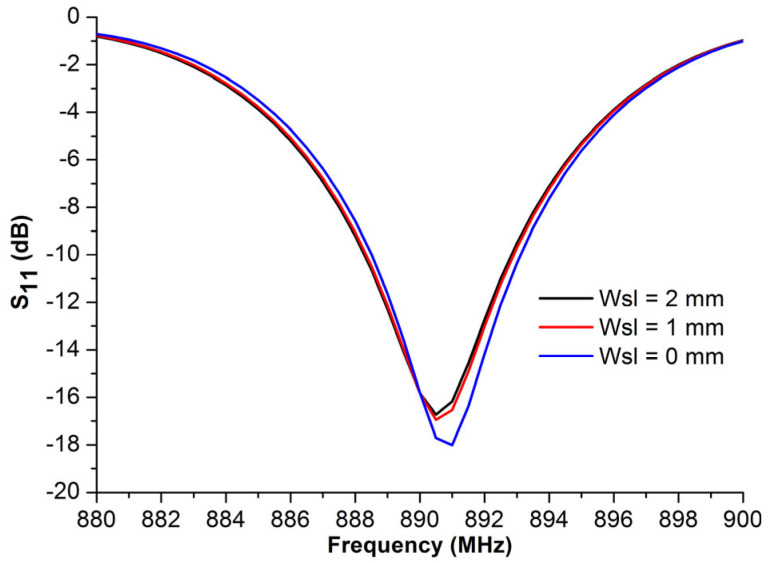
*S*_11_ vs. different slot width values, *W_sl_*.

**Figure 11 micromachines-14-01364-f011:**
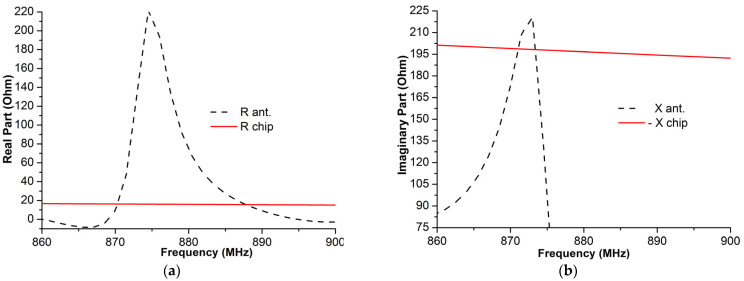
Antenna impedance and the chip impedance in terms of frequency: (**a**) real parts and (**b**) imaginary parts.

**Figure 12 micromachines-14-01364-f012:**
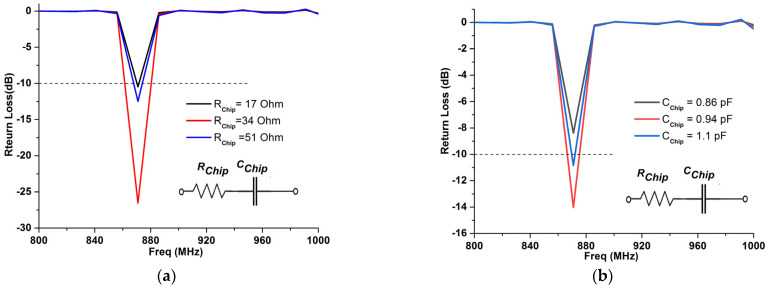
Antenna’s return loss when connected to a chip of *Z_chip_* = *R_C_ + Z_C_*, at 870 MHz. (**a**) *S*_11_ vs. *R_C_*, and (**b**) *S*_11_ vs. *Z_C._*.

**Figure 13 micromachines-14-01364-f013:**
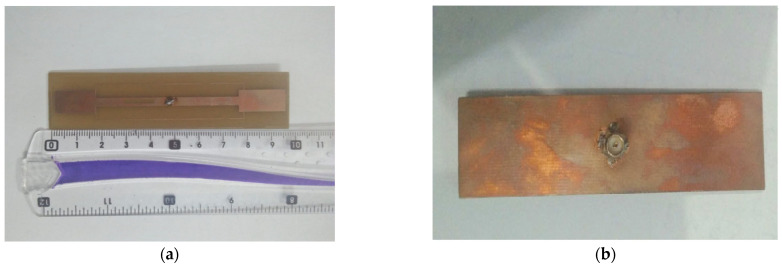
Photograph of the fabricated prototype. Top view (**a**) and bottom view (**b**).

**Figure 14 micromachines-14-01364-f014:**
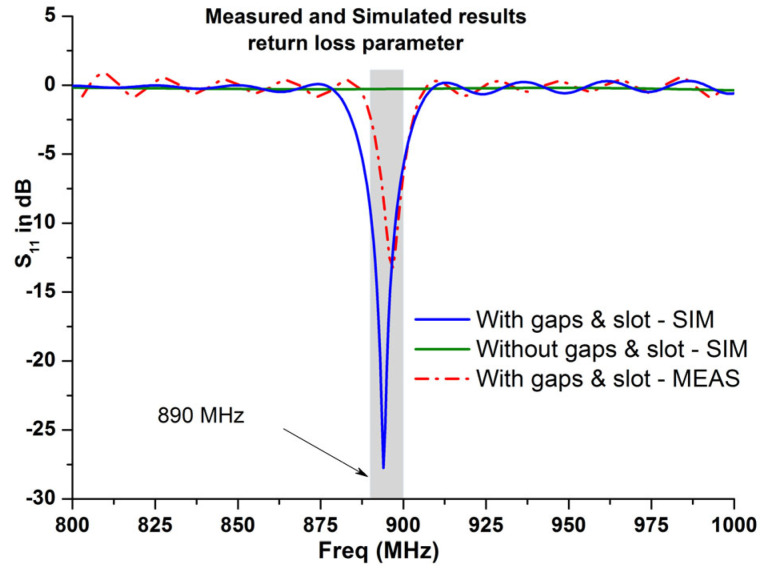
Measured and simulated results of the return loss parameter of the proposed RFID antenna, with and without the gaps and the slot, without considering the industrial chip.

**Figure 15 micromachines-14-01364-f015:**
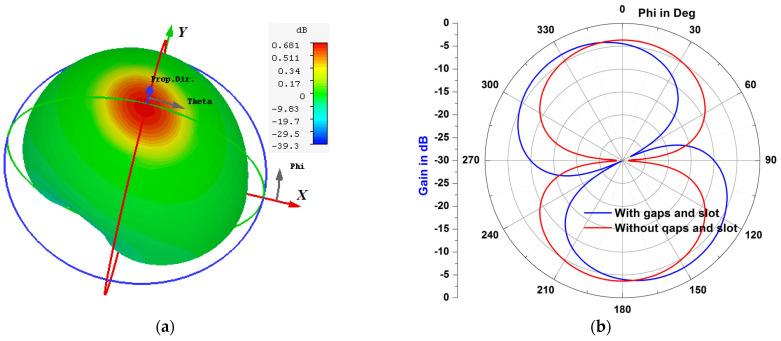
Simulated radiation pattern diagram (gain) of the proposed RFID antenna at 900 MHz: (**a**) 3D radiation in case of the gaps and slot, (**b**) radiation diagram in the XY plane with/without the gaps and the slot.

**Figure 16 micromachines-14-01364-f016:**
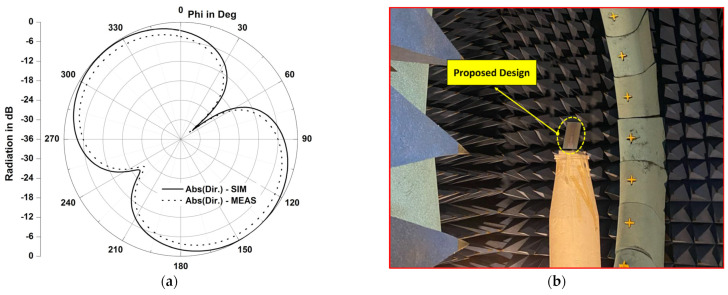
Radiation patter results: (**a**) measured and simulated radiation pattern (directivity) results of the proposed RFID antenna at 900 MHz in the XY plane, (**b**) photograph of measurements process.

**Table 1 micromachines-14-01364-t001:** Chip characteristics [24,28].

Chip	*P_chip_* (dBm)	*R* (Ohm)	*C* (pF)
Impinj Monza5	−17.4	1650	1.21
Alien Higgs	−18	1500	0.825
Impinj M730	−24	2370	0.93

**Table 2 micromachines-14-01364-t002:** Optimized design parameters of the proposed structure.

Symbol	Size (mm)	Symbol	Size (mm)	Symbol	Size (mm)
*L_Sub_*	95	*L* _1_	19	*x* _1_	16.7
*W_Sub_*	24	*W* _1_	10	*x*_2_, *y*_2_	(4, 3.3)
*h*	2	*L* _2_	57.7	*x* _3_	14.3
*L_g_*	2	*W* _2_	4	*x* _4_	5.7
*L_Slot_*	14	*L* _3_	15.7	*y* _4_	1.5
*W_Slot_*	2	*W* _3_	*W* _2_	*W* _4_	0.6

**Table 3 micromachines-14-01364-t003:** The real part and imaginary part with different weight slots, *W_sl_*.

*W_sl_* (mm)	Real Part (Ohm)	Imaginary Part (Ohm)
0	27.6	186.23
1	21.34	185.34
1.5	20.29	184.76
2	16.3	184.123

**Table 4 micromachines-14-01364-t004:** Effect of gap’ length on the read range.

Gap Length (mm)	Antenna Impedance (Ohm)	Read Range (m)	Return Loss at 870.3 MHZ
*lg* = 2	16 − *j*184	17	−11 dB
*lg* = 1.5	133 − *j*229.3	5	−6 dB
*lg* = 1	138 − *j*228	5	−5 dB

**Table 5 micromachines-14-01364-t005:** The read range of several key published papers.

Published Literature	RFID Tag Size (mm^2^)	Read Range (m)	Operating Frequency Band (MHz)
[31]	63 × 24	6	866–952
[32]	83.6 × 51.4	6.8	885
[33]	55.5 × 11.9	5.4	920
[34]	47.1 × 14.8	4.87	915
[35]	36.2 × 8.2	4.1	915
Proposed RFID antenna	*L* × *W*	17	870

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
