# Peer review of "Improved Stable Read Range of the RFID Tag Using Slot Apertures and Capacitive Gaps for Outdoor Localization Applications"

_micromachines, 2023, doi:10.3390/mi14071364_

Round 1

Reviewer 1 Report

The authors present a planar antenna configuration in which layout elements are used to improve the impedance matching with dedicated RFID chips, due to their simple manufacturing and low cost. However, in order for the paper to be published, the authors must solve the following subjects:

 General issues:

·        The Introduction chapter does not compare the proposed antenna matching solution (advantages, drawbacks) with the existing ones, as generally expected. Also, the novelty/originality elements that this work brings are not sufficiently justified. Moreover, slots and gaps defined within antenna elements are often used for frequency tuning or adding resonant behaviour.

·        Please provide the name of software used for 3D electromagnetic simulations, and add all dimensions of metallic routes/elements, drawn in Figure 3, within Table 2.

·        Regarding equation (1), it is desirable to use a single symbol, preferably C instead of c, to designate the capacitor value. Angular frequency should be designated by ω instead of w or W.

·        In (2), the speed of light is usually noted by c.

·        Figure numbers to be changed: Figure 12 (on page 12) becomes Figure 13; Figure 13 becomes Figure 14; Figure 14 becomes Figure 15.

 Technical issues:

·        Equation (2) is valid for point-to-point connections; however, here we are dealing with a backscattering system, whose link budget equations depend on RFID application model. See details in: Griffin, Joshua D., and Gregory D. Durgin. "Complete link budgets for backscatter-radio and RFID systems." IEEE Antennas and Propagation Magazine, vol. 51, no. 2 (2009): pp. 11-25.

·        The calculation of the gap capacitance in a microstrip structure described in Section 3.1 is not correct because the formulas used by the authors, available in [29], refer to the odd-mode capacitance of two coupled microstrip lines. For this reason, rows 137-157 can be removed completely. In addition, the statement "The two metal operates as a capacitor' plates with air as insulator" (see line 146) is not valid, because the dielectric material supporting the antenna must be also considered.

·        Lines 280-281: the authors consider that "It can be seen from Table 3 that a satisfactory impedance matching level is obtained over a wide frequency interval." However, the measured bandwidth does not exceed a few MHz.

·        Lines 290-293: the authors state that "Figure 12 shows clearly that resonant frequency of the UHF RFID tag antenna is around 870 MHz with a return loss less than -10 dB, whereas 3D EM simulations illustrated a resonant frequency around 890 MHz". However, both graphs in Figure 12 show a minimum at the same frequency (870 MHz).

·        New Figure 13.b (antenna’s bottom view) shows the use of a coaxial connector for antenna characterization. Since the antenna requires balanced signal excitation, while the connector is of unbalanced type, the authors have to explain how was solved the compatibility of the connection between these two structures.

·        Rows 311-313: the authors say “Figure 15 (previous 14) shows an omnidirectional case of radiation pattern, hence the read range of proposed RFID antenna will be almost the same over a 180 degree angle variations”. However, the radiation pattern shown in (new) Figure 15 is not omnidirectional (over full 360 degrees)! Adding the antenna position relative to the presented radiation characteristic is therefore required for clarification.

·        The paper states that the proposed layout changes have the effect of improving return loss (RL) to 11dB, but the authors do not present the effect of each layout change on the gain and shape of the antenna directivity characteristic. It is therefore necessary to introduce a graph comparing (i) the initial antenna gain (without layout changes) and (ii) after these changes, and another graph showing the directivity characteristics under the same conditions as mentioned before. It is recommended that the values thus obtained be entered in (2) - (4) for preliminary determination of an ideal link budget.

·        In case of RL = 5dB, the power delivered to the RFID chip is about 69% of the signal power received by the antenna, while for RL = 11dB, the power delivered to the chip reaches 92%. However, increasing the power at the chip input by 33% does not explain an increase in the read range from 5m to 17m.

Note: below are only suggestions for replacing technical terms.

The ususal term is microstrip (not micro-strip); replace it where necessary.

Row 84: replace "conjugate image" with "complex conjugate(d)".

Row 209: replace "metal roof" with "top metal layer of the microstrip structure".

Row 228: replace "well-adapted" with "well-matched".

Author Response

please fimd the attached file for reviewer 1

Reviewer 2 Report

Paper presents a large-read range RFID antenna at UHF band. Also, measuring results are reported. The manuscript is not conceptualized as preferred for the Journal. There are lack of details, introduction, theory of antenna design and analysis and so on. For example we do not know how authors find the initial antenna dimensions. We do not know details of antenna and so on. Authors should better prepare the paper. Authors should address the following items.

1) Authors have been directly proposed the tag configuration in Figure 3. The tag design process is not clear! In other words, what is the origin of the initial guess of the proposed tag configuration?

2) The state of the art analysis should be better analyses in the introduction section and the innovations introduced in this work if there are any should be better detailed.

3) The mentioned equations are some common formulas that are not required to be included in the paper. It seems the authors want to describe some parameters for someone without any knowledge in antenna design.

4) Employing optimization methods in antenna design is not a new topic and I do not see any novelty in this manuscript.

5) The first paragraph of introduction is not related to the manuscript subject.

6) In figure 1, authors mentioned that an equivalent R-C electrical circuit can be considered as a model of RFID chip. Please add suitable reference/references for the sentence.

7) Please check the order of figures.

8) Please highlight the advantages and drawbacks of the proposed antenna. Are there any drawbacks for the proposed antenna?

9) How the numerical results are obtained?

10) Could the authors report the antenna polarization, antenna efficiency, and 3D pattern?

Reviewer 3 Report

This paper discusses a UHF RFID tag antenna and impedance matching approach is used for maximizing the read range. However, authors have not shown how they have obtained the impedance of the chip and the antenna in the simulation. Please incorporate the process / steps to find impedance as it is the key part of this paper and it will be good for better understanding of the readers. 

There are several English spelling and grammar mistakes. So, thoroughly check the whole paper.

Author Response

This paper discusses a UHF RFID tag antenna and impedance matching approach is used for
maximizing the read range. However, authors have not shown how they have obtained the impedance of
the chip and the antenna in the simulation. Please incorporate the process / steps to find impedance as it
is the key part of this paper, and it will be good for better understanding of the readers. 

Author answer:

The impedance of the industrial chip (Impinj M730) is calculated at a specific frequency (870 MHz) using
equation (1) thanks to its resistance R and capacitance C which are given in datasheet (see Table 1). But
antenna impedance is calculated thanks to EM simulation process. During this process we tried to get an
antenna‘ impedance nearer to the complex conjugate of the chip impedance. This was the aim of this
paper, that why we have used capacitive gaps and etched slots techniques to make antenna impedance
approximately the complex conjugate of the chip impedance at 870 MHz.

Reviewer 4 Report

​The design structure parameters in the figure, table and text of the article are not consistent, and one should pay attention to the corresponding cases, italics and subscripts. The drawings of the article are extremely rough, the font and size of the coordinate axes in the figure are inconsistent, the graphics are fuzzy, and some of the drawings are incomplete, as shown in Figure 15 (a). The references are all in the wrong order. We believe that it would be simple and rigorous to standardize the format of the article, but the author has not done so and the overall attitude of the article is not rigorous and serious. I think the author needs to make major revisions to the manuscript before deciding to submit it.

​A revision of the text is recommended.

Author Response

​The design structure parameters in the figure, table and text of the article are not consistent, and one
should pay attention to the corresponding cases, italics and subscripts. The drawings of the article are
extremely rough, the font and size of the coordinate axes in the figure are inconsistent, the graphics are
fuzzy, and some of the drawings are incomplete, as shown in Figure 15 (a). The references are all in the
wrong order. We believe that it would be simple and rigorous to standardize the format of the article, but
the author has not done so and the overall attitude of the article is not rigorous and serious. I think the
author needs to make major revisions to the manuscript before deciding to submit it.

Author answer:
We have to thank dear reviewer for its comment, however we found it very general and, we think, it will be
better if it is more précised. Text font, font size, numbers, figure captions and everything is written
following the rigorous template of the journal. The parts of figure 15-a, which are not shown, are not so
important and one can conclude easily that they are just the parts of Theta and Phi circumferences,
however it has been rectified after Reviewer comment. References are edited using MS Word

References tool and they are cited in desired order upon the need to validate involved ideas following the
order of Sections.
This paper has been revised 15 times before submitting the first version and has been revised again
based on the comments of the two first Reviewers. All graphics have been edited using a professional
graphical tool and we have already published papers where all graphs have been prepared using the
same tool and the same method/template (font type, font size, curve’ lines width, graph’ size (original
size: 8cmx8cm) .. etc.) and we did not receive any comment about them
---------------------------------------------------------------------------------------------------------------------------
[Ref_1]: Pourika Kamalvand,, aurav Kumar Pandey and Manoj Kumar Meshram, “A single-sided
meandered-dual-antenna structure for UHFRFID tags,” in International Journal of Microwave and
Wireless Technologies, 2017, DOI: 10.1017/S1759078716000866

Round 2

Reviewer 1 Report

You have not yet answered a very important technical issue: how you solved the compatibility problem related to (i) the balanced antenna port connected to (ii) the unbalanced connector used for the signal feed; please comment on this.

Reviewer 2 Report

Thanks the authors for answers and updates. Although the quality of the manuscript is improved, However, I think that my concerns are not answered correctly. So, the revised version of the manuscript does not match with the standards of an ISI journal.

Reviewer 4 Report

In the last review, I raised issues such as icon font size and the author considered my review to be general, but we found that the author still did not revise my review. The unit size of the horizontal coordinate of Figures (a) and (b) in Figure 5 is not the same, and the same problem exists in Figures 7, 9, 11, etc. The other coordinates in Figures 8, 10 are S11 but S11 in Figure. Neither of the two red guides in Figure 11 is horizontal. This is only part of the problem with the full text of the graphs, comparing multiple graphs reveals that their font sizes are not consistent. I think these are the most basic issues in the standardization of the article, and it is clear that the author is not serious and is not paying attention to them. The author's reply states that the article diagrams, etc., are strictly in the journal format, but this does not seem to us to be the case.

Author Response

​In the last review, I raised issues such as icon font size and the author considered my review to be general, but we found that the author still did not revise my review. The unit size of the horizontal coordinate of Figures (a) and (b) in Figure 5 is not the same, and the same problem exists in Figures 7, 9, 11, etc. The other coordinates in Figures 8, 10 are S11 but S11 in Figure. Neither of the two red guides in Figure 11 is horizontal. This is only part of the problem with the full text of the graphs, comparing multiple graphs reveals that their font sizes are not consistent. I think these are the most basic issues in the standardization of the article, and it is clear that the author is not serious and is not paying attention to them. The author's reply states that the article diagrams, etc., are strictly in the journal format, but this does not seem to us to be the case

Reviewer 4, 2nd Round: summary of comments:

Dear reviewer, thank you so much for you precious comments that helped us, enormously, to improve the quality of presented graphs and hence the quality of this paper. Here below, in black color, a summary of your comments and suggestions followed with authors ‘answers,

  • The unit size of the horizontal coordinateof Figures (a) and (b) in Figure 5 is not the same, and the same problem exists in Figures 7, 9, 11

Author answer:

The unit size of horizontal axis (x-axis) in each figure where there are two figures, (a) and (b), such in figures 5, 7, 9 and 11, have been modified to be the same in (a) and in (b) (own to each figure): the same Fmin, the same Fmax and the same Step.

In the most graphs we keep the same y-axis configuration (Min value, Max value, and Step), in both (a)- and (b)-figures, but in some ones we couldn’t because it depends on the magnitude of the two quantities plotted in (a)- and (b)-figures.

  • The other coordinates in Figures 8, 10 are S11 but S11 in Figure????:

Author answer:

In the 3rd revised version, x-axis and y-axis of both graphs of figures, 8 and 10, have the same configuration. Graph of figure 4 has been also modified.

  • Neither of the two red guides in Figure 11 is horizontal

Author answer:

They aren’t horizontal lines, they are curves but they seem quasi-horizontal, they represent real and imaginary parts of chip impedance versus frequency

Dear Sir, All graphs have been exported with an original size of 23cm x 17cm, font type, font size and font style of numbers, graph' title, axes' titles, and legends are as follow: Arial, 22 points and Bold for axes’ titles. But when rescaling graphs to a width of 8cm (while keeping width-height ratio) the font size becomes small.

Round 3

Reviewer 2 Report

There is no comment.

Author Response

Thanks for your valuable time